# LARGE LANGUAGE MODEL-ENHANCED MULTI-ARMED BANDITS

## ABSTRACT

Large language models (LLMs) have been adopted to solve sequential decision-making tasks such as multi-armed bandits (MAB), in which an LLM is directly instructed to select the arms to pull in every iteration. However, this paradigm of *direct arm selection* using LLMs has been shown to be suboptimal in many MAB tasks. Therefore, we propose an alternative approach which combines the strengths of classical MAB and LLMs. Specifically, we adopt a classical MAB algorithm as the high-level framework and leverage the strong *in-context learning* capability of LLMs to perform the sub-task of reward prediction. Firstly, we incorporate the LLM-based reward predictor into the classical *Thompson sampling* (TS) algorithm and adopt a decaying schedule for the LLM temperature to ensure a transition from exploration to exploitation. Next, we incorporate the LLM-based reward predictor (with a temperature of 0) into a *regression oracle*-based MAB algorithm equipped with an explicit exploration mechanism. We also extend our TS-based algorithm to *dueling bandits* where only the preference feedback between pairs of arms is available, which requires non-trivial algorithmic modifications. We firstly conduct empirical evaluations on synthetic MAB tasks, where the results show that our algorithms consistently outperform LLM-based direct arm selection. Additionally, we perform experiments using real-world text datasets, in which the results demonstrate that in challenging tasks where the arms lack semantic meanings that can be exploited by the LLM, our approach delivers significantly better performance than LLM-based direct arm selection.

## 1 INTRODUCTION

Large language models (LLMs) have demonstrated impressive capabilities in various tasks Liu et al. (2024a); OpenAI (2023a;b). As a result, many recent works have leveraged LLMs as agents to solve real-world sequential decision-making tasks. Specifically, some recent works have adopted powerful pre-trained LLMs to solve *multi-armed bandit* (MAB) problems Chen et al. (2024); Krishnamurthy et al. (2024); Mukherjee et al. (2024); Xia et al. (2024). These works usually directly instruct a pre-trained LLM to select the next arm to pull and do not require the costly LLM fine-tuning. However, this paradigm has been demonstrated to lead to sub-optimal MAB algorithms in many scenarios Krishnamurthy et al. (2024). Specifically, it has been observed that directly using an LLM for arm selection often struggles to explore efficiently in real-world environments. To this end, we propose an alternative paradigm which combines classical MAB algorithms with LLMs such that we can *achieve the best of both worlds*. Specifically, we leverage a classical MAB algorithm as the high-level framework, and adopt a pre-trained LLM (without fine-tuning) to perform the sub-task of *reward prediction* based on the history of (the features of) the selected arms and their observed rewards. Compared to the previous approach of directly employing an LLM for arm selection Krishnamurthy et al. (2024), this allows us to *leverage the strength of LLMs in in-context learning* (ICL) to solve prediction (i.e., supervised learning) tasks. In other words, instead of using an LLM to replace the MAB algorithm, we **leverage LLMs to enhance classical MAB algorithms**.

We further motivate our approach by drawing analogy to recent works aiming to improve the performance of LLMs in complex reasoning tasks via tree search methods Bi et al. (2024); Hao et al. (2023); Yao et al. (2024); Zhang et al. (2024). Specifically, these methods often adopt a classical tree search algorithm as the high-level framework (e.g., Monte-Carlo tree search), and use LLMs to perform different sub-tasks such as *reward/value prediction*, action generation, etc. Therefore,

their overall paradigm aligns with our approach of *using classical algorithms to guide the high-level decision-making while leveraging the strengths of LLMs in performing some sub-tasks*. For example, the work of Koh et al. (2024) has also used a pre-trained LLM for reward prediction based on the past history to improve classical algorithms. Specifically, they have adopted best-first search as the high-level reasoning framework in web automation and used a pre-trained multimodal LLM as a reward/value function in the framework. Additionally, Liu et al. (2024b) adopted a similar approach, focusing on single-step decision making. Specifically, they proposed leveraging a pre-trained LLM to derive a utility function, which is then maximized to guide action selection.

In order to incorporate an LLM as a reward predictor into MAB in a principled way, we adopt two classical MAB algorithms as our high-level framework which are naturally amenable to the integration of an LLM-based reward predictor. Firstly, we adopt the classical Thompson sampling (TS) algorithm Thompson (1933) and use a powerful pre-trained LLM to sample the reward values used in TS, hence introducing our *Thompson Sampling with LLM* (TS-LLM) algorithm. We ensure a proper balance between exploration and exploitation by carefully controlling the temperature of the LLM. That is, we ensure that the temperature is large enough in the initial stages to achieve sufficient exploration and gradually decay its value to promote more exploitation in later stages. Secondly, we adopt a *regression oracle*-based MAB algorithm Foster & Rakhlin (2020) and leverage the LLM as the regression oracle for reward prediction, to introduce our *Regression Oracle-based bandit with LLM* (RO-LLM). Since the algorithm from Foster & Rakhlin (2020) is equipped with an explicit exploration mechanism and hence only needs the LLM to provide an accurate reward prediction, we set the LLM temperature to 0 to remove the randomness in the reward prediction.

In addition to classical stochastic MAB, we also introduce an LLM-enhanced algorithm for *dueling bandits* Li et al. (2024); Verma et al. (2024); Yue et al. (2012). In dueling bandits, instead of a single arm, a pair of arms are selected in every iteration, after which a *binary preference observation* is revealed indicating which arm is preferred over the other. Thanks to the prevalence of preference feedback, dueling bandits are widely applicable in various important real-world scenarios, such as recommender systems Yang et al. (2024c), alignment of LLMs (via reinforcement learning from human feedback) Dwaracherla et al. (2024), among others. However, adapting our algorithms to dueling bandits is non-trivial due to the need to handle preference feedback (rather than numerical feedback) and to select a pair of arms. We adapt our TS-LLM algorithm discussed above to introduce the *Thompson Sampling with LLM for Dueling Bandits* (TS-LLM-DB) algorithm. In order to achieve a seamless integration of the LLM (as a reward predictor) into dueling bandits, we have leveraged the theoretical equivalence between the maximizers of the *Borda function* and the latent reward function in dueling bandits Mehta et al. (2023) (more details in Sec. 3.3).

Note that in addition to the strong reward prediction capability of LLMs, another benefit of our LLM-enhanced MAB algorithms is that they do not require us to specify the form of the unknown reward function. Specifically, classical MAB algorithms are usually only able to handle a specific class of reward functions, such as linear reward functions Abbasi-Yadkori et al. (2011). As a result, misspecification of the reward function (i.e., when the groundtruth reward function does not lie in the pre-specified function class) has been an important challenge in MAB, and many efforts have been made to address this difficulty Ghosh et al. (2017); Wang et al. (2024b). In contrast, due to the flexibility of LLMs to predict reward functions of varying degrees of complexity, our algorithms can automatically adapt to the level of difficulty of the problem. As a result, we are free from the requirement to specify the class of reward functions beforehand.

We use extensive experiments to demonstrate the empirical advantage of our algorithms. We firstly use synthetic stochastic MAB experiments to show that our TS-LLM and RO-LLM algorithms both consistently outperform baseline methods which directly instruct the LLM to select actions (Sec. 4.1). Next, we show that our TS-LLM-DB algorithm achieves small regrets in synthetic dueling bandit experiments (Sec. 4.2). We also apply our TS-LLM to contextual MAB experiments designed using two real-world text datasets (Sec. 4.3). The results show that in the experiments where the LLM can exploit the semantic meanings of the arm features (to accurately predict the association between the contexts and arms), directly instructing the LLM to select actions leads to strong performance which is comparable to our TS-LLM. In the more challenging tasks in our experiments where the arms lack such semantic information, LLM-based direct arm selection suffers from significant performance degradation, and our TS-LLM performs dramatically better. We expect our findings to provide useful and practical guidelines for future works and applications adopting LLMs as agents to solve real-world sequential decision-making tasks.

## 2 PROBLEM SETTING

**Multi-Armed Bandits (MAB).** In our problem setting, every arm $i = 1, \ldots, K$ is associated with a $d$-dimensional feature vector $x_i \in \mathbb{R}^d$ and the reward of an arm $i$ is a function of its feature vector $x_i$: $f(x_i)$. For example, in the classical linear bandits, the reward of arm $i$ is given by a linear function: $f(x_i) = \theta^\top x_i$ with an unknown $\theta$. In every iteration $t$, an MAB algorithm selects an arm $i_t$ to pull, and observes a corresponding noisy reward $y_t = f(x_{i_t}) + \epsilon$ where $\epsilon$ is usually a zero-mean Gaussian noise. The goal of an MAB algorithm is usually to minimize the *cumulative regret*: $R_T = \sum_{t=1}^{T} [f(x_{i^*}) - f(x_{i_t})]$ where $i^* = \arg\max_{i=1,\ldots,K} f(x_i)$ represents the optimal arm. We also consider the setting of contextual bandits (Sec. 4.3), in which in every iteration $t$, we receive a new set of $K$ arms denoted as $\mathcal{I}_t = \{i_1^t, \ldots, i_K^t\}$ and choose an arm $i_t$ from $\mathcal{I}_t$. When selecting an arm in iteration $t$, an MAB algorithm needs to make use of (the feature vectors of) the previously selected arms and their corresponding rewards: $\mathcal{D}_{t-1} = \{(x_{i_s}, r_s)\}_{s=1,\ldots,t-1}$. Therefore, we will include $\mathcal{D}_{t-1}$ in the prompt for the LLM-based agent in our algorithms.

**Dueling Bandits.** In dueling bandits, in every iteration $t$, we select a pair of arms $i_{t,1}$ and $i_{t,2}$ and observe binary preference feedback $r_t = \mathbb{1}(i_{t,1} \succ i_{t,2})$, which is equal to 1 if $i_{t,1}$ is preferred over $i_{t,2}$ and 0 otherwise. We assume that the preference observation $r_t$ is generated by the commonly adopted BTL model Hunter (2004); Luce (2005). Specifically, there exists a latent reward function $f$ which maps the feature vector $x_i$ of an arm $i$ to its corresponding latent reward value $f(x_i)$. For a pair of arms $i_{t,1}$ and $i_{t,2}$, the preference probability (i.e., the probability that arm $i_{t,1}$ is preferred over arm $i_{t,2}$) under the BTL model is given by $\mathbb{P}(i_{t,1} \succ i_{t,2}) = \mu(f(x_{i_{t,1}}) - f(x_{i_{t,2}}))$, in which $\mu : \mathbb{R} \to [0,1]$ is the logistic function: $\mu(z) = 1/(1 + e^{-z})$. The preference observation $r_t = \mathbb{1}(i_{t,1} \succ i_{t,2})$ is then assumed to be sampled from a Bernoulli distribution with the probability $\mathbb{P}(i_{t,1} \succ i_{t,2})$. A common notion of regret in dueling bandits is $R_T = \sum_{t=1}^{T} [2f(x_{i^*}) - f(x_{i_{t,1}}) - f(x_{i_{t,2}})]$. However, in practical applications, we usually need to devise a method to recommend an arm during the dueling bandit algorithm Lin et al. (2024). Our LLM-based algorithm for dueling bandits recommends the first selected arm $i_{t,1}$ as the best arm (more details in Sec. 3.3). Therefore, in our experiments (Sec. 4.2), we report the regret of the first arm: $R_T = \sum_{t=1}^{T} [f(x_{i^*}) - f(x_{i_{t,1}})]$, which we think is more relevant in practice.

## 3 LLM-ENHANCED MAB ALGORITHMS

### 3.1 THOMPSON SAMPLING WITH LLM (`TS-LLM`)

Our `TS-LLM` (Algo. 1) employs the LLM to predict the reward of every arm and leverages *the inherent randomness in the LLM-generated text* to achieve exploration. Specifically, in every iteration $t$, we include the current history of observations $\mathcal{D}_{t-1} = \{x_{i_s}, r_s\}_{s=1,\ldots,t-1}$ in the prompt for the LLM. For each arm $i = 1, \ldots, K$, we append its feature vector $x_i$ to the end of the prompt and instruct the LLM to predict its reward $\hat{r}_{t,i}$ (line 3 of Algo. 1). The prompt adopted in this step is shown in App. C.1. Then, the arm with the largest predicted reward $\hat{r}_{t,i}$ is selected (line 5 of Algo. 1).

To achieve a gradual transition from exploration to exploitation, we choose a schedule for the temperature of the LLM which *decays across iterations*. As a result, at the initial stage when significant exploration is required, we use a large temperature to induce sufficient randomness in the LLM-generated reward prediction. In later stages when a larger degree of exploitation is more beneficial, we use a small temperature to reduce the randomness in the reward prediction. This allows us to naturally combine the powerful reward prediction form the LLM, thanks to its impressive in-context learning (ICL) capability, and the classical TS algorithm to derive a coherent algorithm for arm selection in MAB. We empirically verify that such a decaying schedule of temperatures indeed achieves better performance than using a fixed temperature in Sec. 5.1.

**Justifications for `TS-LLM` (Algo. 1).** Our `TS-LLM` algorithm shares a similar motivation with some previous works which have also relied on the randomness in the output generated by the LLM to achieve exploration in sequential decision-making tasks. For example, the work of Yang et al. (2024b) has adopted an LLM with a large temperature to select a batch of diverse input queries for Bayesian optimization; the work of Liu et al. (2024c) has used an LLM to predict the performance achieved by different hyperparameter configurations in Bayesian optimization, and used the variance of multiple independently sampled predictions from the LLM as the exploration term in their upper confidence bound-based algorithm. Another line of works with similar underlying principles as our

---

**Algorithm 1** `TS-LLM`

---
1: **for** iteration $t = 1, \ldots, T$ **do**
2:    **for** arm $i = 1, \ldots, K$ **do**
3:       $\widehat{r}_{t,i} = \text{LLM}(\mathcal{D}_{t-1}, x_i)$ // predict reward
4:    Select arm $i_t = \arg\max_{i=1,\ldots,K} \widehat{r}_{t,i}$, observe reward $r_t$
5:    Update history $\mathcal{D}_t = \mathcal{D}_{t-1} \cup \{(x_{i_t}, r_t)\}$

---

**Algorithm 2** `RO-LLM`

---
1: **for** iteration $t = 1, \ldots, T$ **do**
2:    **for** arm $i = 1, \ldots, K$ **do**
3:       $\widehat{l}_{t,i} = \text{LLM}(\mathcal{D}_{t-1}, x_i)$ // predict loss
4:    Let $j_t = \arg\min_{i=1,\ldots,K} \widehat{l}_{t,i}$
5:    **for** arm $i = 1, \ldots, K$ and $i \neq j_t$ **do**
6:       $p_{t,i} = \frac{1}{\mu + \gamma(\widehat{l}_{t,i} - \widehat{l}_{t,j_t})}$
7:    Let $p_{t,j_t} = 1 - \sum_{i \neq j_t} p_{t,i}$
8:    Sample $i_t \sim p_t$, observe loss $l_t$ (negated reward)
9:    Update history $\mathcal{D}_t = \mathcal{D}_{t-1} \cup \{(i_t, l_t)\}$

---

`TS-LLM` is approximating Thompson sampling (TS) with neural networks. Some previous works have adopted an ensemble of neural networks (NNs) Osband et al. (2016; 2023); Dwaracherla et al. (2024) and performed approximate TS by randomly sampling from the ensemble. In contrast, we approximate the posterior reward distribution in TS using the stochastic predictions generated by the LLM in our `TS-LLM` algorithm.

### 3.2 REGRESSION ORACLE-BASED MAB WITH LLM (`RO-LLM`)

A recent line of works have proposed to adopt a generic *regression oracle* for reward prediction in MAB, and incorporated explicit exploration mechanisms to derive theoretically principled algorithms Foster et al. (2018); Foster & Rakhlin (2020). Interestingly, the high-level principle of these works aligns well with our approach in this work, i.e., adopting a model capable of reward prediction (i.e., a regression oracle in these previous works and an LLM in our work) and utilizing a separate high-level framework to achieve exploration. Therefore, here we incorporate an LLM as the regression oracle into the `SquareCB` algorithm from Foster & Rakhlin (2020), hence proposing our `RO-LLM` algorithm (Algo. 2). To be consistent with Foster & Rakhlin (2020), instead of rewards, we consider the observations as *losses* (line 10 of Algo. 2), which are simply the negation of rewards.

In every iteration $t$ of our `RO-LLM` algorithm, we use the LLM to predict the loss $\widehat{l}_{t,i}$ of every arm $i$ (line 3 of Algo. 2). Here we adopt the same prompt as the `TS-LLM` algorithm (shown in App. C.1), except that here we use losses as observations rather than rewards. Next, we choose the arm with the smallest predicted loss and denote it as $j_t$ (line 4). After that, we use the LLM-based loss predictions to construct a distribution $p_t$ over all $K$ arms (lines 5-7), from which the next arm $i_t$ is sampled (line 8). Note that the `SquareCB` algorithm from Foster & Rakhlin (2020) is equipped with an explicit exploration mechanism (via the sampling distribution $p_t$). As a result, unlike our `TS-LLM` algorithm, here we no longer need to exploit the inherent randomness in the LLM-generated output to achieve exploration. Therefore, when using the LLM for loss prediction in our `RO-LLM` algorithm (line 3 of Algo. 2), we set the temperature of the LLM to 0 and hence obtain deterministic reward predictions.

### 3.3 THOMPSON SAMPLING WITH LLM FOR DUELING BANDITS (`TS-LLM-DB`)

Here we introduce our `TS-LLM-DB` algorithm for dueling bandit, in which we select a pair of arms $i_{t,1}$ and $i_{t,2}$ in every iteration and collect a binary observation indicating their relative preference $r_t = \mathbb{1}(i_{t,1} \succ i_{t,2})$.

**Preference Probability Prediction.** In contrast to our `TS-LLM` (Algo. 1) and `RO-LLM` (Algo. 2) which use an LLM to predict the *reward* of every arm, our `TS-LLM-DB` algorithm (Algo. 3) instead adopts an LLM to predict the *probability that an arm is preferred over another arm*. Specifically, when adopting the LLM for preference probability prediction via ICL (line 4 of Algo. 3), for the $s^{\text{th}}$

---

**Algorithm 3** `TS-LLM-DB`

---

1: **for** iteration $t = 1, \ldots, T$ **do**
2:      **for** arm $i = 1, \ldots, K$ **do**
3:          **for** uniformly sampled arm $j = 1, \ldots, N$ **do**
4:              $\widehat{p}_{t,i,j} = \text{LLM}(\mathcal{D}_{t-1}, [x_i, x_j])$
5:          Calculate $\widehat{r}_{t,i} = \frac{1}{N} \sum_{n=1}^{N} \widehat{p}_{t,i,n}$
6:      Select the first arm $i_{t,1} = \arg\max_{i=1,\ldots,K} \widehat{r}_{t,i}$
7:      **for** arm $j = 1, \ldots, K$ **do**
8:          $\widehat{p}_{t,j} = \text{LLM}(\mathcal{D}_{t-1}, [x_j, x_{i_{t,1}}])$
9:      Select the second arm $i_{t,2} = \arg\max_{i=1,\ldots,K} \widehat{p}_{t,i}$
10:     Observe binary preference $r_t = \mathbb{1}(i_{t,1} \succ i_{t,2})$
11:     Update history $\mathcal{D}_t = \mathcal{D}_{t-1} \cup \{([i_{t,1}, i_{t,2}], r_t)\}$

---

input-output pair in the dataset $\mathcal{D}_{t-1}$ included in the prompt, the input corresponds to *the features of the pair of arms $x_{i_{s,1}}$ and $x_{i_{s,2}}$* (instead of a single arm in Algo. 1 and Algo. 2). The corresponding output represents the observed preference $r_s = \mathbb{1}(i_{s,1} \succ i_{s,2})$. When predicting the preference probability of a pair of arms $x_i$ and $x_j$, we append their features at the end of the prompt, denoted as $[x_i, x_j]$ (line 4 of Algo. 3). As a result, the LLM is able to *predict the probability that the first arm $x_i$ is preferred over the second arm $x_j$*, i.e., predict $\mathbb{P}(x_i \succ x_j)$. The prompt template we have adopted here is shown in App. C.1.

**Representing The Features of Arm Pairs.** We adopt two approaches to incorporate the features of a pair of arms into the prompt. Firstly, when the latent reward function $f$ is linear: $f(x) = \theta^\top x$, we have that $\mathbb{P}(x_1 \succ x_2) = \mu(f(x_1) - f(x_2)) = \mu\left(\theta^\top (x_1 - x_2)\right)$. That is, the preference probability $\mathbb{P}(x_1 \succ x_2)$ is a function of the difference $x_1 - x_2$. Therefore, we use the difference between the feature vectors of the first arm and second arm (i.e., $x_1 - x_2$) in the prompt. Secondly, when the latent reward function is non-linear, the preference probability is no longer a function of $x_1 - x_2$. In this case, we concatenate the feature vectors of $x_1$ and $x_2$ and include them in the prompt.

**Selection of A Pair of Arms.** To select the pair of arms $i_{t,1}$ and $i_{t,2}$ in every iteration, we draw inspirations from the arm selection strategy from the work of Verma et al. (2024). Specifically, in iteration $t$, for every arm $i$, we use the LLM to predict the probability that arm $i$ is preferred over $N$ uniformly sampled arms and calculate their average predicted probability $\widehat{r}_{t,i}$ (line 3-5 of Algo. 3). Then, we choose the first arm by maximizing $\widehat{r}_{t,i}$ (line 6). This is equivalent to approximately maximizing the *Borda function* $f_{\text{borda}}$ Xu et al. (2020), which is defined as the expected probability that an arm is preferred over a randomly selected arm: $f_{\text{borda}}(x) = \mathbb{E}_{j \in \mathcal{U}([K])}[\mathbb{P}(x \succ x_j)]$ where $\mathcal{U}([K])$ denotes the uniform distribution among all $K$ arms. Specifically, we estimate the expectation in $f_{\text{borda}}(x)$ by uniformly and independently sampling $N$ arms (lines 3-5 of Algo. 3). Theoretically, maximizing the Borda function $f_{\text{borda}}$ is equivalent to maximizing the latent reward function $f$ (see Sec. 2) Mehta et al. (2023). Therefore, the first arm $i_{t,1}$ is selected greedily, i.e., via pure exploitation. As a result, after each iteration $t$, we let our `TS-LLM-DB` algorithm recommend the first arm as the best arm. To choose the second arm, we firstly predict the probability that each arm is preferred over the first arm $i_{t,1}$ (lines 7-8 of Algo. 3), and then select the second arm by maximizing this predicted probability (line 9). This is inspired by the TS-based algorithm from Verma et al. (2024), which encourages the second selected arm to both have large reward and be different from $i_{t,1}$ and all previously selected arms. The work of Verma et al. (2024) has theoretically shown that such an approach to selecting the pair of arms lead to strong performances (i.e., small cumulative regrets) both in theory and in practice.

## 4 EXPERIMENTS

We firstly apply our `TS-LLM` and `RO-LLM` algorithms to synthetic stochastic MAB tasks with both linear and non-linear reward functions (Sec. 4.1). Next, we apply our `TS-LLM-DB` algorithm to solve synthetic dueling bandit problems (Sec. 4.2). Lastly, we adopt contextual bandit tasks designed using two real-world text datasets (Sec. 4.3) to unveil interesting insights about our algorithms. We adopt GPT-3.5-Turbo OpenAI (2023a) as the black-box LLM in the majority of our experiments, and also use DeepSeek-V3 Liu et al. (2024a), Llama-3.3-70B-Instruct Grattafiori et al. (2024) and Qwen2.5-72B-Instruct Yang et al. (2024a) in the experiments in Sec. 4.3.

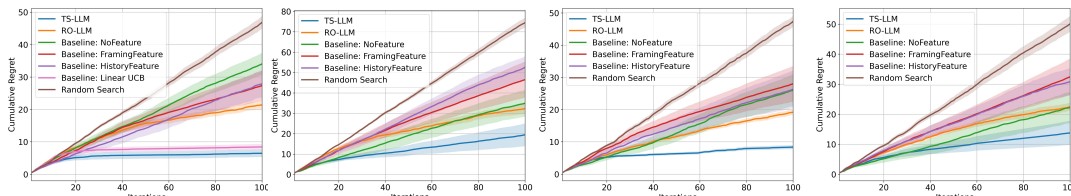

Linear Reward Function    Square Reward Function    Sinusoidal Reward Function    Function Sampled from GP

Figure 1: The performance of our `TS-LLM` and `RO-LLM` in classical stochastic MAB tasks.

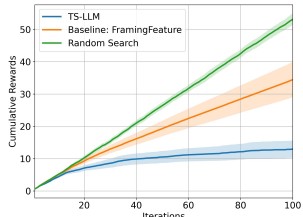

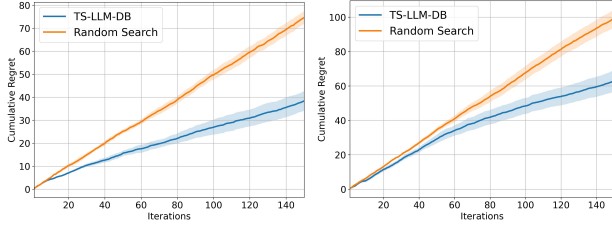

Figure 2: The performance of our `TS-LLM` with linear reward function and $K = 50$ arms.

Figure 3: The performance of our `TS-LLM-DB` in dueling bandits with linear and square latent reward functions.

## 4.1 `TS-LLM` AND `RO-LLM` FOR CLASSICAL STOCHASTIC MAB

We adopt 4 different reward functions in the stochastic MAB experiments here: a linear function, a square function, a sinusoidal function and a function sampled from a Gaussian process (GP). Every arm is associated with a $d = 4$-dimensional feature vector, and we use $K = 16$ arms in all experiments here. We compare our `TS-LLM` and `RO-LLM` algorithms with some baseline algorithms from the work of Krishnamurthy et al. (2024). Specifically, we adopt the best prompt design from Krishnamurthy et al. (2024), i.e., the prompt design which achieved the largest median reward among a total of 32 prompt designs when using GPT-3.5 in the hard bandit instance. Notably, this prompt design *utilizes Chain-of-Thought (CoT) prompting*, as detailed in the prompts provided in Appendix C.3. The prompt designs from Krishnamurthy et al. (2024) do not take into account the features of the arms, therefore, we have proposed and tested multiple variants of their baseline algorithm which differ in terms of the position of the arm features: (a) *Baseline NoFeature*: the original algorithm from Krishnamurthy et al. (2024); (b) *Baseline FramingFeature*: we add the arm features after the problem framing; (c) *Baseline HistoryFeature*: we add the arm features immediately before the history of interactions. In addition, in the experiment with linear reward function, we have also compared with *the classical baseline of Linear UCB*. The cumulative regrets of different methods are shown in Fig. 1, which demonstrate that both our `TS-LLM` and `RO-LLM` achieve smaller regrets than the baseline from Krishnamurthy et al. (2024). Moreover, for linear reward function, the classical Linear UCB algorithm achieves smaller regrets than all LLM-based baseline methods, as well as our `RO-LLM`. However, remarkably, our `TS-LLM` achieves comparable performance to Linear UCB. In addition, Fig. 1 shows that our `TS-LLM` significantly outperforms our `RO-LLM` algorithm, which is likely attributed to the strong exploration capability enabled by the inherent randomness in the LLM-generated output (Sec. 3.1). On the other hand, our `RO-LLM` algorithm generally have smaller variance across multiple trials, which is indicated by the narrower error bars. This is likely due to the use of a temperature of 0 in our `RO-LLM` algorithm (Sec. 3.2) and may make our `RO-LLM` algorithm more desirable in scenarios where more consistent performance is preferred. In addition, we have also tested the performance of our `TS-LLM` with a larger number of arms ($K = 50$). The results in Fig. 2 demonstrate the robustness of the performance advantage of our `TS-LLM`.

## 4.2 DUELING BANDITS

Here we apply our `TS-LLM-DB` algorithm to solve dueling bandit problems with two different latent reward functions $f$: a linear function and a square function. Same as the experiments in Sec. 4.1, we also let $d = 4$ and $K = 16$. In our experiments here, when selecting the first arm, we use $N = 15$ uniformly sampled arms to approximate the Borda function (Sec. 3.3). Similar to the experiments

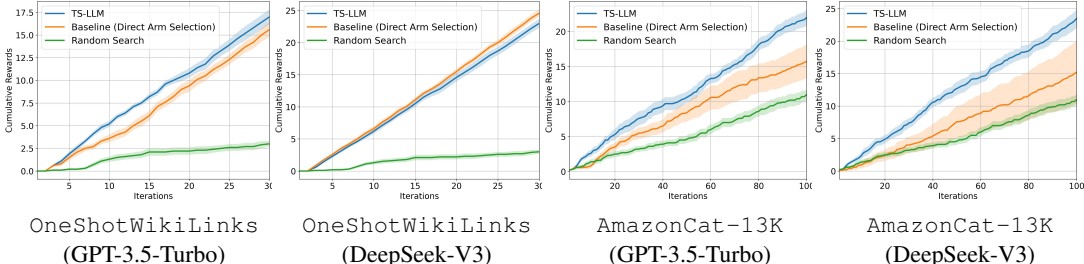

| OneShotWikiLinks | OneShotWikiLinks | AmazonCat-13K | AmazonCat-13K |
| (GPT-3.5-Turbo) | (DeepSeek-V3) | (GPT-3.5-Turbo) | (DeepSeek-V3) |

Figure 4: The cumulative rewards in the text experiments using the `OneShotWikiLinks` and `AmazonCat-13K` datasets (Sec. 4.3). Higher values indicate better performance.

on classical stochastic MAB (Sec. 4.1), we also adopt a decaying schedule of temperature when selecting both arms. Since our `TS-LLM-DB` selects the first arm greedily (i.e., pure exploitation) and chooses the second arm optimistically by balancing exploration and exploitation (Sec. 3.3), we adopt a schedule of smaller temperatures when selecting the first arm to encourage exploitation. As we have discussed in Sec. 3.3, for the linear latent reward function, we use the difference between the feature vectors of the first arm and the second arm as the feature vector in the prompt; for the non-linear square function, we instead adopt the concatenation of the pair of feature vectors.

The results are shown in Fig. 3. Following the common practice in dueling bandits Lin et al. (2024); Verma et al. (2024), here we have reported the reward of the first selected arm (i.e., $f(x_{i_{t,1}})$) in every iteration $t$. This is because the first arm is selected to be the one that is predicted to achieve the largest reward (Sec. 3.3). Here we have only compared with the baseline of random search, because it is highly non-trivial to adapt the algorithm from Krishnamurthy et al. (2024) to the sophisticated dueling bandit problem. As shown in the figures, our `TS-LLM-DB` significantly outperforms random search for both reward functions. Moreover, the regrets are generally larger in the more challenging problem of non-linear (square) reward function.

### 4.3 REAL-WORLD DATASETS WITH TEXT FEATURES

Here we perform experiments using two real-world text dataset: the `OneShotWikiLinks` dataset Singh et al. (2012); Vasnetsov (2018) and the `AmazonCat-13K` dataset Bhatia et al. (2016), both of which have been widely adopted in previous works on contextual bandits Chen et al. (2024). The `OneShotWikiLinks` dataset Singh et al. (2012); Vasnetsov (2018) is a named-entity recognition task in which the contexts consist of text phrases surrounding the mention text (both preceding and following it), and *the arms are text phrases* representing concept names. `AmazonCat-13K` Bhatia et al. (2016) is an extreme multi-label dataset where the contexts are text phrases derived from the title and content of an item, and *the arms are integers* representing item tags. Thus, in the former dataset, the arm features (i.e., the text phrases) contain semantic information that is likely beneficial for the LLM in selecting arms, whereas in the latter dataset, the arm features lack such semantic content. As a result, the latter dataset (i.e., `AmazonCat-13K`) *requires a larger degree of exploration* and is hence more challenging.

We apply our `TS-LLM` to the tasks here, because it achieves smaller regrets than `RO-LLM` in the synthetic experiments (Sec. 4.1). Since it is non-trivial to adapt the method from Krishnamurthy et al. (2024) to the sophisticated problem setting here, we instead compare our `TS-LLM` with a baseline which is obtained by modifying the prompt of our algorithm (originally designed for reward prediction) to instead directly select an arm. We refer to this baseline method as *Baseline (Direct Arm Selection)*. We have included the prompt templates used by our `TS-LLM` and the baseline (for both experiments) in App. C.4. We consider $K = 10$ randomly sampled arms (i.e., 10 concept names in `OneShotWikiLinks` and 10 items in `AmazonCat-13K`) in the experiments, and adopt two powerful black-box LLMs: GPT-3.5-Turbo and DeepSeek-V3.

The results are shown in Fig. 4. The figures show that our `TS-LLM` algorithm achieves comparable performance with the baseline of direct arm selection in the `OneShotWikiLinks` task and *significantly outperforms the baseline in the `AmazonCat-13K` task*. This is likely because in `OneShotWikiLinks` task, the powerful LLMs possesses in-depth knowledge about the semantic meanings of the individual arms, i.e., the names of the entities. As a result, given some context (i.e., the text before and after the entity), the LLM is able to accurately choose the corresponding

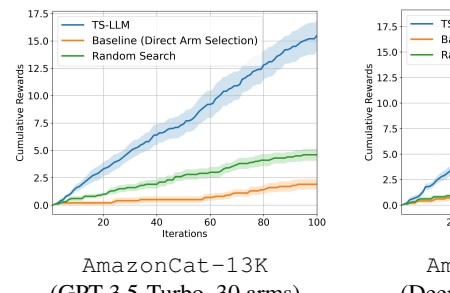 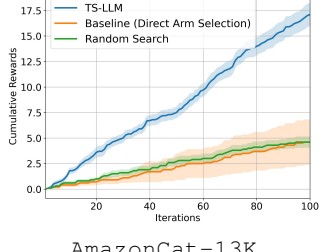

AmazonCat-13K
(GPT-3.5-Turbo, 30 arms)

AmazonCat-13K
(DeepSeek-V3, 30 arms)

Figure 5: The cumulative rewards in the text experiments using the AmazonCat-13K dataset with $K = 30$ arms.

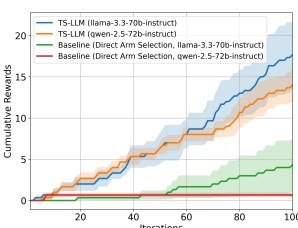 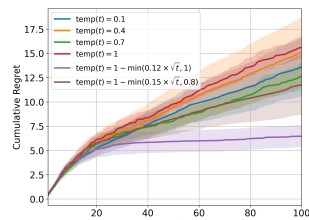

Figure 6: Results using two other LLMs: Llama-3.3-70B-Instruct and Qwen2.5-72B-Instruct (AmazonCat-13K).

Figure 7: The performance of our TS-LLM algorithm in stochastic MAB tasks with different temperatures.

arm whose semantic meaning is associated with the context, which explains the strong performance of the baseline of direct arm selection in the OneShotWikiLinks task. On the other hand, in the AmazonCat-13K task, since the arms lack such semantic information useful for the LLMs, the LLMs are not able to accurately infer the association between the contexts (i.e., text phrases describing an item) and the arms (i.e., integers representing item tags). Therefore, in such tasks, an algorithm *needs to perform substantial exploration* in order to learn the association between the contexts and the arms and hence to achieve small regrets. The inadequate performance of the baseline algorithm in this task can likely be attributed to *the inability of LLM-based direct arm selection to engage in efficient exploration*, which aligns with the findings from Krishnamurthy et al. (2024). Meanwhile, thanks to *the strong exploration capability of the high-level classical TS mechanism* (Sec. 3.1), our TS-LLM algorithm is able to efficiently explore the space of arms and hence to achieve small regrets in this task.

To further verify this insight, we have additionally conducted an experiment using the AmazonCat-13K dataset in a more challenging setting, i.e., with a larger number of arms (i.e., $K = 30$). The results (Fig. 5) show that the performance advantage of our TS-LLM over the baseline is further enlarged. Therefore, the results in Figs. 4 and 5 demonstrate that compared with the approach of directly instructing the LLM to select arms, **our TS-LLM algorithm is particularly beneficial in challenging tasks where considerable exploration is required**. On the other hand, LLM-based direct arm selection is expected to perform well in scenarios where the LLM has significant knowledge about the arms or the association between the contexts and the arms.

**Robustness of Our Findings Across Different LLMs.** We have also conducted experiments using the AmazonCat-13K dataset with two additional LLMs: Llama-3.3-70B-Instruct Grattafiori et al. (2024) and Qwen2.5-72B-Instruct Yang et al. (2024a). The results in Fig. 6 demonstrate that the performance advantage of our TS-LLM is consistent across various LLMs.

## 5 ABLATION STUDY

### 5.1 IMPACT OF DIFFERENT TEMPERATURES

Here we investigate the impact of the temperature of the LLM on the performance of our TS-LLM (Algo. 1). As we have discussed in Sec. 3.1, we adopt a decaying schedule for the LLM temperature to ensure a transition from exploration to exploitation. We follow the same experimen-

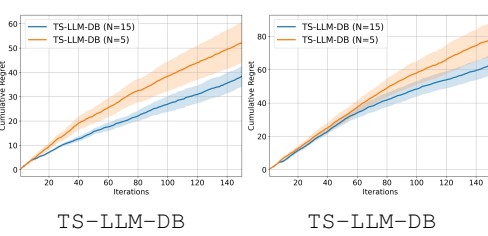

| TS-LLM-DB | TS-LLM-DB |
|:---:|:---:|
| (Linear Reward Function) | (Square Reward Function) |

Figure 8: Impact of the number of uniformly sampled arms $N$ when estimating the Borda function to select the first arm in TS-LLM-DB.

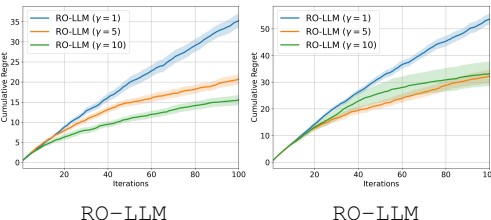

| RO-LLM | RO-LLM |
|:---:|:---:|
| (Linear Reward Function) | (Square Reward Function) |

Figure 9: The impact of the exploration parameter $\gamma$ in our RO-LLM algorithm.

tal setting as Sec. 4.1 and adopt the linear reward function. The results in Fig. 7 show that the best performance is achieved by adopting decaying LLM temperatures, whereas fixing the temperature to various values leads to inferior performance. This is because fixing the temperature to a large value hinders the exploitation capability of TS-LLM in later stages, while the use of a fixed small temperature results in insufficient exploration in the initial stage.

## 5.2 Impact of the Number of Samples $N$ When Selecting the First Arm in TS-LLM-DB

Recall that our TS-LLM-DB algorithm selects the first arm by approximately maximizing the Borda function $f_{\text{borda}}$ (Sec. 3.3), in which we use $N$ randomly sampled arms to approximate the expectation in $f_{\text{borda}}$ (lines 3-5 of Algo. 3). Fig. 8 presents the results of our TS-LLM-DB with different values of $N$, which demonstrate that a larger $N$ improves the performance because it leads to a better approximation of $f_{\text{borda}}$. However, also note that the use of a larger $N$ increases the number of API calls to the LLM and hence incurs more cost. Therefore, in practice, the value of $N$ should be selected based on the trade-off between the desired performance and the budget.

## 5.3 Impact of The Exploration Parameter in Our RO-LLM Algorithm

The parameter $\gamma$ in our RO-LLM can be used to control the degree of exploration. As can be seen from lines 4-7 of Algo. 2, a larger value of $\gamma$ results in a larger weight (in the arm sampling distribution $p_t$) on the arm $j_t$ that is predicted to be the best arm. Therefore, a larger $\gamma$ leads to greater emphasis on *exploitation*, thereby reducing the focus on *exploration*. Here we test three values of $\gamma$ and display the results in Fig 9. The figures show that an overly small value of $\gamma = 1$ significantly deteriorates the performance due to excessive exploration. In the relatively simpler MAB problem with a linear reward function, a larger $\gamma = 10$ (i.e., more emphasis on exploitation) benefits the algorithm since only minimal exploration is required to learn the simple reward function. Meanwhile, in the more difficult problem with a non-linear (square) function, a larger $\gamma = 10$ leads to worse regrets than $\gamma = 5$ since a larger degree of exploration is needed compared to the linear function.

## 6 Conclusion

In this work, we propose an alternative paradigm of LLM-based sequential decision making and focus on the MAB problem. We adopt a classical MAB algorithm as the high-level framework and leverage the strong in-context learning capability of LLMs to perform the sub-task of reward prediction. Synthetic experiments show that our algorithms consistently outperform baseline methods of LLM-based direct arm selection. Through contextual MAB experiments designed using two real-world text datasets, we show that in challenging tasks where the arm features are not associated with semantic meanings exploitable by the LLM, our TS-LLM performs significantly better than LLM-based direct arm selection.

## REPRODUCIBILITY STATEMENT

To ensure reproducibility, we have clearly described our detailed experimental setting in Sec. 4 and App. C, which are necessary to reproduce our experimental results. The prompts adopted by our algorithms are all listed in App. C.

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

## A    RELATED WORK

**LLM-Based Multi-Armed Bandits (MAB).** The work of Krishnamurthy et al. (2024) used an LLM to sequentially choose the arms in MAB. They have consider standard MAB problems with a finite number of arms, and their results have shown that LLMs struggle in MAB tasks in most scenarios (i.e., for most of their prompt designs). Some recent works proposed techniques to improve the exploration capability of LLMs in MAB tasks, such as random subsampling from the interaction history Monea et al. (2024), and adding summary statistics of the interaction history to the prompt Nie et al. (2024). The work of Harris & Slivkins (2025) demonstrated that LLMs can be used to accelerate the exploration in MAB tasks by selecting promising candidates from a large action space. The work of Chen et al. (2024) proposed to adopt an LLM-based arm selection strategy in the initial stage of MAB and gradually switch to classical MAB algorithms in later stages. However, their method requires the availability of the likelihood of the LLMs and are hence not able to adopt the typically more powerful black-box LLMs such as ChatGPT. The work of Xia et al. (2024) proposed an LLM-based algorithm for dueling bandits. Compared with our `TS-LLM-DB` (Sec. 3.3), they have considered a simpler setting of dueling bandits in which the preference feedback is generated by a preference matrix. In contrast, we have adopted the BTL model (Sec. 2), which allows us to take into account the arm features and hence makes our setting more general. Mukherjee et al. (2024) proposed to train a decision transformer to predict the rewards of different arms in MAB and hence to assist in arm selection.

**Other LLM-Based Sequential Decision-Making Methods.** In addition to MAB, some previous works have proposed methods to incorporate LLMs into other sequential decision-making algorithms. For example, some prior works have used LLMs to improve the performance of Bayesian optimization (BO) by either directly instructing the LLM to sequentially select the input queries in BO Yang et al. (2024b) or using LLMs to enhance different components of BO (such as initial input selection, surrogate model prediction, etc.) Liu et al. (2024c). A number of recent works have used the transformer model to learn a policy for action selection in reinforcement learning Dai et al. (2024); Laskin et al. (2022); Lee et al. (2024). The field of LLM-based agents is broad and has garnered significant attention due to the rapidly advancing capabilities of modern LLMs. Many surveys on LLM-based agents have been released Cheng et al. (2024); Wang et al. (2024a); Xi et al. (2023), offering comprehensive overviews of this area.

## B    STATEMENT OF LLM USAGE

We have used LLMs to help polish the writing of the paper.

## C    MORE EXPERIMENTAL DETAILS AND RESULTS

In all our synthetic experiments (Secs. 4.1 and 4.2), the MAB tasks have $K = 16$ features and the feature vectors of the arms are $4$-dimensional. All our experiments are conducted using a computer with an 11th Gen Intel Core i7-1165G7 4-Core Processor and 16GB RAM. The `OneShotWikiLinks` dataset Singh et al. (2012); Vasnetsov (2018) and the `AmazonCat-13K` dataset Bhatia et al. (2016) are under the CC BY 4.0 license. All error bars in the plots represent the standard error of the mean.

### C.1    THE PROMPT TEMPLATE ADOPTED BY OUR ALGORITHMS

Below is the prompt we have used for our `TS-LLM` algorithm (Algo. 1) and `RO-LLM` algorithm (Algo. 2) in classical stochastic bandits experiment in Sec. 4.1. Here every [INPUT] contains the feature vectors of an arm, and every [OUTPUT] corresponds to its corresponding observed reward.

> **Prompt for Our `TS-LLM` and `RO-LLM`**
>
> Help me predict the function value at the last input. Each function value is associated with a Normal distribution with a fixed but unknown mean. Your response should only contain the function value in the format of #function value#.
> input: [INPUT], output: [OUTPUT]
> input: [INPUT], output: [OUTPUT]
> ...
> input: [INPUT], output:

The template below is the prompt we have used for our `TS-LLM-DB` algorithm (Algo. 3) in the dueling bandit experiment in Sec. 4.2. Here every [INPUT] contains the difference or concatenation of the feature vectors of a pair of arms (see Sec. 4.2 for more details), and every [OUTPUT] corresponds to a binary observation which is equal to 1 if the first arm is preferred over the second arm and 0 otherwise. Although the output labels for each data point in the prompt is binary, here we have instructed the LLM to predict a continuous value, to ensure that the LLM-generated output can be used as the preference probability.

> **Prompt for Our `TS-LLM-DB`**
>
> Help me predict the value for the last input as a continuous value between 0 and 1. Your response MUST only contain the value in the format of #value#.
> input: [INPUT], output: [OUTPUT]
> input: [INPUT], output: [OUTPUT]
> ...
> input: [INPUT], output:

### C.2 MORE DETAILS ON THE SYNTHETIC EXPERIMENTS (SECS. 4.1 AND 4.2)

In our synthetic experiments in Sec. 4.1, we have adopted synthetic functions as the reward functions $f$, including linear function: $f(x) = \theta^\top x$, square function: $f(x) = (\theta^\top x)^2$, sinusoidal function: $f(x) = \sin(\theta^\top x)$, and a function sampled from a Gaussian process with a length scale of 0.4. We repeat each experiment 10 times with a different random seed for each repetition. We run each method for 100 iterations, with the initial 2 arms randomly selected. We add a Gaussian noise with a noise variance of 0.02 to each observation. In all our experiments here, we have adopted the optimal schedule for the temperature discovered in Sec. 5.1 (Fig. 7). For the classical baseline of Linear UCB in the experiment with linear reward function, an extensive grid search was conducted for the exploration parameter. The search range was from 0.01 to 1 with a step size of 0.01. Additionally, the groundtruth observation noise variance (i.e., 0.02) was used as the regularization parameter $\lambda$.

In our synthetic experiments on dueling bandits in Sec. 4.2, we adopt the following latent reward functions: linear function: $f(x) = \theta^\top x$, and square function: $f(x) = (\theta^\top x)^2$. We repeat each experiment 5 times with a different random seed for each repetition. We run each method for 150 iterations, with the initial 2 arms randomly selected. As we have discussed in Sec. 4.2, we use a decaying schedule of LLM temperatures, and adopt a smaller schedule of temperatures when selecting the first arm to encourage exploitation. Specifically, for linear latent reward function, in iteration $t$, we use $\text{temp}(t) = 1.5 - \min(0.1 \times \sqrt{t}, 1.4)$ as the temperature when selecting the first arm and use $\text{temp}(t) = 1.5 - \min(0.1 \times \sqrt{t}, 1.1)$ when choosing the second arm. For the square latent reward function, we adopt larger values of the temperature, because the non-linear reward function makes the dueling bandit problem more challenging and hence a larger degree of exploration is needed. Specifically, we use $\text{temp}(t) = 1.6 - \min(0.13 \times \sqrt{t}, 1.5)$ when choosing the first arm and let $\text{temp}(t) = 1.6 - \min(0.13 \times \sqrt{t}, 1.1)$ when selecting the second arm.

In our experiments here, we use the BTL model to obtain the preference observation (Sec. 2). Specifically, after a pair of arms $i_{t,1}$ and $i_{t,2}$ are selected, we firstly calculate their preference probability:

$$\mathbb{P}(x_{i_{t,1}} \succ x_{i_{t,2}}) = \frac{1}{1 + e^{-10(f(x_{i_{t,1}}) - f(x_{i_{t,2}}))}}. \tag{1}$$

We have added a 10 in the exponent to reduce the noise in the preference observations and hence simplify the dueling bandit problem. Then, we sample the binary reward observation $r_t$ from a Bernoulli distribution with the probability $\mathbb{P}(x_{i_{t,1}} \succ x_{i_{t,2}})$.

## C.3 More Details about the Baseline Algorithms

Here we present the prompts we have used for different baseline algorithms we have used in Sec. 4.1. Specifically, how we have modified the prompt from the LLM-based MAB method from Krishnamurthy et al. (2024) in different ways in order to incorporate the features of the arms, to make their method comparable with our algorithms. We have highlighted the arm features we have added in blue.

---

**Baseline: NoFeature**

You are in a room with 16 buttons labeled
['blue', 'green', 'red', 'yellow', 'purple', 'orange', 'cyan', 'magenta', 'lime', 'pink', 'teal', 'lavender', 'brown', 'beige', 'maroon', 'mint']
Each button is associated with a Normal distribution with a fixed but unknown mean; the means for the buttons could be different and are associated with features of buttons. For each button, when you press it, you will get a reward that is sampled from the button's associated distribution.
You have 100 time steps and, on each time step, you can choose any button and receive the reward. Your goal is to maximize the total reward over the 100 time steps. So far you have played [TIMES] times with the following choices and rewards:
[COLOR] button, reward [REWARD]
[COLOR] button, reward [REWARD]
...
You MUST output a distribution over the 16 buttons as probabilities, formatted EXACTLY like this example: #[COLOR]:p1,[COLOR]:p2,...,[COLOR]:p16#. Each probability value(p1,p2,...,p16) MUST be a number between 0 and 1, and the total of all probabilities MUST equal 1.
Let's think step by step to make sure we make a good choice. Which button will you choose next? YOU MUST provide your final answer within the tags <Answer>DIST </Answer>where DIST is #[COLOR]:p1,[COLOR]:p2,...,[COLOR]:p16#.

---

**Baseline: FramingFeature**

You are in a room with 16 buttons labeled
['blue', 'green', 'red', 'yellow', 'purple', 'orange', 'cyan', 'magenta', 'lime', 'pink', 'teal', 'lavender', 'brown', 'beige', 'maroon', 'mint']
Feature of [COLOR] button: [FEATURE]
Feature of [COLOR] button: [FEATURE]
...
Each button is associated with a Normal distribution with a fixed but unknown mean; the means for the buttons could be different and are associated with features of buttons. For each button, when you press it, you will get a reward that is sampled from the button's associated distribution.
You have 100 time steps and, on each time step, you can choose any button and receive the reward. Your goal is to maximize the total reward over the 100 time steps. So far you have played [TIMES] times with the following choices and rewards:
[COLOR] button, reward [REWARD]
[COLOR] button, reward [REWARD]
...
You MUST output a distribution over the 16 buttons as probabilities, formatted EXACTLY like this example: #[COLOR]:p1,[COLOR]:p2,...,[COLOR]:p16#. Each probability value(p1,p2,...,p16) MUST be a number between 0 and 1, and the total of all probabilities MUST equal 1.
Let's think step by step to make sure we make a good choice. Which button will you choose next? YOU MUST provide your final answer within the tags <Answer>DIST </Answer>where DIST is #[COLOR]:p1,[COLOR]:p2,...,[COLOR]:p16#.

---

**Baseline: HistoryFeature**

You are in a room with 16 buttons labeled
['blue', 'green', 'red', 'yellow', 'purple', 'orange', 'cyan', 'magenta', 'lime', 'pink', 'teal', 'lavender', 'brown', 'beige', 'maroon', 'mint']
Each button is associated with a Normal distribution with a fixed but unknown mean; the means for the buttons could be different and are associated with features of buttons. For each button, when you press it, you will get a reward that is sampled from the button's associated distribution.
You have 100 time steps and, on each time step, you can choose any button and receive the reward. Your goal is to maximize the total reward over the 100 time steps.
Feature of [COLOR] button: [FEATURE]
Feature of [COLOR] button: [FEATURE]
...
So far you have played [TIMES] times with the following choices and rewards:
[COLOR] button, reward [REWARD]
[COLOR] button, reward [REWARD]
...
You MUST output a distribution over the 16 buttons as probabilities, formatted EXACTLY like this example:
#[COLOR]:p1,[COLOR]:p2,...,[COLOR]:p16#. Each probability value(p1,p2,...,p16) MUST be a number between 0 and 1, and the total of all probabilities MUST equal 1.
Let's think step by step to make sure we make a good choice. Which button will you choose next? YOU MUST provide your final answer within the tags <Answer>DIST </Answer>where DIST is #[COLOR]:p1,[COLOR]:p2,...,[COLOR]:p16#.

---

## C.4 MORE DETAILS ON THE TEXT EXPERIMENTS

Here we present more details on the experiment in Sec. 4.3 in which we have adopted a real-world text dataset. Every experiment in this section is repeated 10 times with a different random seed in every repetition, except that the experiments in Fig. 6 are repeated 3 times.

In the experiment using the `OneShotWikiLinks` dataset, contexts exceeding 400 words were first removed. Then, 10 concept names were randomly selected, each associated with 2,000 to 3,000 contexts. Finally, 2,000 contexts were randomly sampled for each of these 10 concept names. For the experiment using the `AmazonCat-13K` dataset, contexts exceeding 500 characters in length were first removed. Then, only data containing a single item tag was retained. Finally, the top 10 or 30 item tags with the highest number of contexts were selected, and all corresponding data were used as experimental data. The number of data samples for the 10-arm and 30-arm experiments were 34,227 and 40,287, respectively.

We display below the prompts we have used for our `TS-LLM` algorithm and the baseline algorithm in the two text datasets. For fair comparisons, we keep most of the contents between the prompts of the two methods identical. Therefore, the only major difference between the prompts of the two methods is that the prompt for the baseline method directly instructs the LLM to select the next arm to pull. On the other hand, in the prompt for our `TS-LLM` algorithm, we let the LLM predict the score of a combination of a context and an arm. As a result, the prompt for our `TS-LLM` bears a larger degree of resemblance to standard in-context learning, because we are effectively leveraging the LLM to solve a supervised learning task.

## Prompt for `TS-LLM` in OneShotWikiLinks task

**Task Description**
At the TEST DATA, Please assign a reward indicating how well the Incomplete Text aligns with the Previous Text and Next Text.

**reward**:
- 0 indicates poor alignment.
- 1 indicates perfect alignment.
- A reward closer to 1 should only be assigned when the Incomplete Text is perfectly aligned with the surrounding texts.

**The Incomplete Text can be one of the following words**:
['Microsoft Windows', 'Telugu', 'XML', 'Moscow', 'help', 'MTV', 'Halloween', 'Ottoman Empire', 'Soviet', 'Bangladesh'].

The reward value MUST be a number between 0 and 1. Your response MUST be the reward value only, formatted as #reward value#.

Below are previous examples:
**Previous Text**: [PREVIOUS TEXT]
**Next Text**: [NEXT TEXT]
**Incomplete Text**: [INCOMPLETE TEXT]
**Reward**: [REWARD]

**Previous Text**: [PREVIOUS TEXT]
**Next Text**: [NEXT TEXT]
**Incomplete Text**: [INCOMPLETE TEXT]
**Reward**: [REWARD]

...

###TEST DATA:
This is the TEST DATA for which the reward needs to be assigned:
**Previous Text**: [PREVIOUS TEXT]
**Next Text**: [NEXT TEXT]
**Incomplete Text**: [INCOMPLETE TEXT]
**Reward**:

---

### Prompt for the Baseline Method in OneShotWikiLinks task

The task is to choose the most suitable word to complete the Incomplete Text from the following list of options in order to earn the most reward:
['Microsoft Windows', 'Telugu', 'XML', 'Moscow', 'help', 'MTV', 'Halloween', 'Ottoman Empire', 'Soviet', 'Bangladesh'].
Your response MUST only contain one word from the list.

Reward indicates how well the Incomplete Text aligns with the Previous Text and Next Text.
- 0 indicates poor alignment.
- 1 indicates perfect alignment.

Below is the historical data:
**Previous Text**: [PREVIOUS TEXT]
**Next Text**: [NEXT TEXT]
**Incomplete Text**: [INCOMPLETE TEXT]
**Reward**: [REWARD]

**Previous Text**: [PREVIOUS TEXT]
**Next Text**: [NEXT TEXT]
**Incomplete Text**: [INCOMPLETE TEXT]
**Reward**: [REWARD]

...

Below is the incomplete text for which you need to complete:
**Previous Text**: [PREVIOUS TEXT]
**Next Text**: [NEXT TEXT]
**Incomplete Text**:

---

### Prompt for `TS-LLM` in AmazonCat task

There are Titles and Contents of some items.

Labels and items correspond one-to-one.
There are a total of 10 items.The Labels MUST be ONE of the following numbers: [2571, 1471, 7961, 12246, 5754, 342, 5456, 5960, 11235, 10688]

The Reward is a number between 0 and 1 determined by whether the Label is correct or not.

Help me predict the Reward at the last Title, Content and Label.

Your response MUST be the predicted Reward only, formatted as #predicted Reward#.

**Title**: [Title]
**Content**: [Content]
**Label**: [Label]
**Reward**: [REWARD]

**Title**: [Title]
**Content**: [Content]
**Label**: [Label]
**Reward**: [REWARD]

...

**Title**: [Title]
**Content**: [Content]
**Label**: [Label]
**Reward**:

### Prompt for the Baseline Method in AmazonCat task

There are Titles and Contents of some items.

Labels and items correspond one-to-one.
There are a total of 10 items.The Labels MUST be ONE of the following numbers: [2571, 1471, 7961, 12246, 5754, 342, 5456, 5960, 11235, 10688]

The Reward is a number between 0 and 1 determined by whether the Label is correct or not.

Help me choose the correct Label at the last Title and Content.  Your response MUST be the chosen Label only, formatted as #chosen Label#.

**Title**: [Title]
**Content**: [Content]
**Label**: [Label]
**Reward**: [REWARD]

**Title**: [Title]
**Content**: [Content]
**Label**: [Label]
**Reward**: [REWARD]

...

**Title**: [Title]
**Content**: [Content]
**Label**:

