# OpenReview forum: "Large Language Model-Enhanced Multi-Armed Bandits"
_ICLR.cc/2026/Conference — ICLR 2026 Conference Withdrawn Submission_

### Official Review · Reviewer_kvtf · 2025-10-28

**Soundness:** 2
**Presentation:** 2
**Contribution:** 2
**Rating:** 2
**Confidence:** 4

**Summary:**

The paper proposes a general paradigm: keep a classical bandit algorithm for decision-making and delegate reward prediction to an LLM. Concretely: (i) TS-LLM uses temperature-decayed sampling as a proxy for Thompson Sampling; (ii) RO-LLM plugs an LLM as a regression oracle within SquareCB with explicit exploration; (iii) TS-LLM-DB handles dueling bandits by having the LLM estimate pairwise preference probabilities and then approximately maximizing Borda scores for the first arm while balancing explore/exploit for the second. Synthetic tasks and two contextual text setups show that when arm semantics are weak, the proposed methods substantially outperform “LLM-direct-selection,” and TS-LLM competes with Linear-UCB on linear rewards.

**Strengths:**

- The paper proposes an intuitive paradigm that integrates classical MAB exploration mechanisms with LLM-based reward prediction.


- The experiments cover diverse settings and demonstrate consistent gains over baselines across multiple LLMs, enhancing the generality of the findings.


- The approach is plug-and-play, requiring no LLM fine-tuning, and the paper offers insights that are directly applicable to practitioners using black-box LLM APIs.

**Weaknesses:**

- Although the paper repeatedly motivates the use of Thompson Sampling and SquareCB, it does not provide any regret or convergence analysis—both of which are fundamental in the bandit literature. Even a simplified approximation or asymptotic argument would meaningfully strengthen the claim of a “principled integration.”




- The evaluation primarily compares against prompt-engineering variants from Krishnamurthy et al. (2024), omitting many strong and relevant bandit methods as well as transformer-based reward predictors. In addition, some experiments rely only on Random Search as the baseline, which is too weak to demonstrate the superiority of the proposed LLM-enhanced approach. Without stronger comparisons or ablation studies, it remains unclear which component of the method actually contributes to performance gains.


- The paper equates LLM temperature-induced randomness with posterior sampling in Thompson Sampling but does not justify this equivalence. It is unclear how increasing the temperature systematically corresponds to more exploration—especially since higher temperatures can also yield incoherent or irrelevant generations rather than structured exploration of uncertainty.


- The feature dimension d=4 used in experiments appears small, raising questions about whether the proposed method can generalize to higher-dimensional or more complex feature spaces where distinguishing between arms becomes harder.


- The paper lacks analysis of computational and monetary costs, including API usage, latency, and sensitivity to prompt phrasing or context length. Since each arm evaluation requires an LLM call, scalability and efficiency are major concerns. It remains unclear whether the use of an LLM offers sufficient benefits over classical or learned bandit algorithms to justify its overhead.

**Questions:**

- The central question concerns the claimed LLM-enhanced nature of the approach. Without comprehensive comparisons against strong non-LLM predictors, it remains unclear whether the “enhancement” truly stems from LLM capabilities. Could the authors clarify what unique contribution the LLM provides beyond standard function approximators?


- What are the practical benefits of using LLM-based reward predictors compared to classical or neural regressors (e.g., XGBoost, GLMs, or lightweight MLP heads) under identical bandit wrappers? In what real-world scenarios would this hybrid design be preferable or even necessary?


- How sensitive is performance to prompt phrasing versus numeric hyperparameters such as the temperature schedule or exploration constants? Has any systematic prompt ablation or rephrasing been conducted to assess robustness?


- Can the authors provide a diagnostic or heuristic for deciding when to use direct LLM arm selection versus the proposed hybrid (e.g., based on arm semantic richness, task structure, or LLM calibration quality)?

---

### Official Review · Reviewer_puob · 2025-10-29

**Soundness:** 3
**Presentation:** 3
**Contribution:** 2
**Rating:** 4
**Confidence:** 3

**Summary:**

The paradigm of direct arm selection using LLMs has been shown to be suboptimal in many multi-armed bandit (MAB) tasks. This paper proposes an alternative approach which combines the strengths of classical MAB and LLMs. Specifically, the authors adopt a classical MAB algorithm as the high-level framework and leverage the strong in-context learning capability of LLMs to perform the sub-task of reward prediction. Firstly, the authors incorporate the LLM-based reward predictor into the classical Thompson sampling (TS) algorithm and adopt a decaying schedule for the LLM temperature to ensure a transition from exploration to exploitation. Next, the authors incorporate the LLM-based reward predictor (with a temperature of 0) into a regression oracle-based MAB algorithm equipped with an explicit exploration mechanism. The authors also extend the proposed TS-based algorithm to dueling bandits where only the preference feedback between pairs of arms is available, which requires non-trivial algorithmic modifications. The authors firstly conduct empirical evaluations on synthetic MAB tasks, where the results show that the proposed algorithms consistently outperform LLM-based direct arm selection. Additionally, the authors perform experiments using real-world text datasets, in which the results demonstrate that in challenging tasks where the arms lack semantic meanings that can be exploited by the LLM, our approach delivers significantly better performance than LLM-based direct arm selection.

**Strengths:**

1. The studied problem, applying LLMs in sequential decision-making tasks, is very interesting.
2. The authors propose a Thompson sampling (TS)-based algorithm which incorporates the LLM-based reward predictor into the classical TS algorithm, and further extend the proposed TS-based algorithm to dueling bandits which only use the preference feedback between pairs of arms.
3. This paper provides extensive experiments to demonstrate the empirical advantage of the proposed algorithms compared to the baseline algorithm of direct arm selection using LLMs.

**Weaknesses:**

1. There is no theoretical analysis provided in this paper.
2. This paper looks more like an experimental report of applying LLMs to the MAB tasks. The innovation in algorithm design and theoretical analysis is limited. In addition, the significance of the proposed algorithms is also not very clear, since they seem to be designed only for the application of LLMs in MAB tasks (which is an interesting but very particular problem) and compared only with direct arm selection using LLMs. Can the proposed algorithms outperform traditional MAB algorithms, e.g., TS and UCB? It would enhance the paper if the authors can elaborate more on the significance of the proposed algorithms.

**Questions:**

Please see the weaknesses above.

---

### Official Review · Reviewer_Jxss · 2025-10-31

**Soundness:** 3
**Presentation:** 3
**Contribution:** 2
**Rating:** 6
**Confidence:** 2

**Summary:**

The paper proposes using LLMs to enhance classical multi-armed bandit (MAB) algorithms where the LLM acts as a reward predictor, and exploration and decision-making are handled by standard bandit machinery. Specifically, the authors introduced two variants: TS-LLM, which plugs LLM reward predictions into Thompson sampling and uses a decaying temperature schedule to move from exploration to exploitation, and RO-LLM, which treats the LLM as a regression oracle for deterministic predictions. The approach is further extended to dueling bandits (TS-LLM-DB) via Borda-function maximization with LLM-predicted pairwise preference probabilities. Experiments on synthetic MABs and contextual tasks seem to show that the proposed methods consistently outperform direct LLM arm selections.

**Strengths:**

+ It’s interesting to see an algorithm that integrates an LLM-based reward predictor into the classical Thompson sampling framework, which indeed manages the exploration-exploitation trade-off by starting with a high LLM temperature and gradually decaying it to promote exploitation as more data is gathered.

+ To my knowledge, the design of TS-LLM-DB adaptation is novel, which predicts pairwise preferences, approximates the Borda score to pick the first arm, and then selects a second arm to balance novelty and quality.

+The empirical evaluation seems to be comprehensive with both synthetic and real-world data, and it seems to show that existing LLM-based agents are heavily reliant on semantic shortcuts, where the proposed method can address the short-coming.

**Weaknesses:**

- My first concern is the computational cost. The proposed methods (TS-LLM and RO-LLM) seem to require $K$ (the number of arms) separate LLM calls per iteration to get a predicted reward for each arm. The dueling bandit version (TS-LLM-DB) is even more expensive, requiring $(K * N) + K$ LLM calls per iteration. I wonder if this would be prohibitive for many real-world applications.

- In addition, it seems that the algorithms feed the entire history of observations into the LLM prompt for in-context learning. It would be better if the authors could address the case when the number of iterations $t$ grows large.

- Finally, I’m also curious about the prompt sensitivity: The entire system hinges on the LLM's ability to understand the ICL prompt for reward prediction. While the authors provide their prompts in the appendix, I wonder if the performance is sensitive to this prompt's formatting.

**Questions:**

Please refer to my summary of weaknesses.

---

### Official Review · Reviewer_aj7v · 2025-11-03

**Soundness:** 3
**Presentation:** 3
**Contribution:** 2
**Rating:** 2
**Confidence:** 3

**Summary:**

The paper introduces a hybrid approach for the multi-armed bandit (MAB) setting in which they retain a classical bandit algorithm (such as Thompson Sampling) as the high-level decision framework, while leveraging a large language model (LLM) as a reward predictor submodule. Specifically, the LLM is used to predict the expected reward of pulling each arm given the history and context; the bandit algorithm uses those predictions to select actions. Empirical evaluations on both synthetic and real-text‐based tasks show that this “LLM + classical bandit” method outperforms prior pure-LLM direct-arm-selection baselines, especially in cases where the arm options lack intuitive semantic meaning (where an LLM alone struggles).

**Strengths:**

The paper studies the application of LLMs in classical bandit problems, which is an interesting adaptation.

The proposed methods borrowed ideas from bandits research (i.e., Thompson sampling and SquareCB), as scaffolding to facilitate high-level decision-making, while LLMs perform low-level predictions. From this perspective, the designs are theoretically grounded.

Some experiments are conducted, showing that with the proposed scaffolding, the LLM + bandits approach is performing better than just LLM, which is an interesting observation, showing that suitable framework can help LLM in decision-making tasks.

**Weaknesses:**

- The overall message (or the purpose) of this paper is not very clear. If the target is to provide a scaffolding for LLM in decision-making tasks, more experiments should be performed (especially in real-world scenarios and may be not only bandit settings). It seems that the current target is only to help LLM play bandits, which is a very narrative topic and does not provide a sufficient amount of interests from my perspective.

- If only doing LLM play bandits, there should be also more approaches to be tested. For example, the current approach is LLM performing prediction and bandit algorithms performing action selection. Another natural idea is to have several regression algorithms performing a set of predictions and related statistics, and LLM doing the high-level action selection. Also, it can be tried to have LLM performing prediction first and then LLM selecting actions based on the prediction.

Overall, I would suggest expanding the focus of the paper and sharing more insights, either from bandits side or for LLM development.

**Questions:**

My major concerns are listed in weakness.

---

### Note · Authors · 2026-01-06

I have read and agree with the venue's withdrawal policy on behalf of myself and my co-authors.